# Beyond 1-WL with Local Ego-Network Encodings

**Nurudin Alvarez-Gonzalez**[*]
Universitat Pompeu Fabra
nuralgon@gmail.com

**Andreas Kaltenbrunner**
ISI Foundation
Universitat Pompeu Fabra
kaltenbrunner@gmail.com

**Vicenç Gómez**
Universitat Pompeu Fabra
vicen.gomez@upf.edu

## Abstract

Identifying similar network structures is key to capture graph isomorphisms and learn representations that exploit structural information encoded in graph data. This work shows that ego-networks can produce a structural encoding scheme for arbitrary graphs with greater expressivity than the Weisfeiler-Lehman (1-WL) test. We introduce IGEL, a preprocessing step to produce features that augment node representations by encoding ego-networks into sparse vectors that enrich Message Passing (MP) Graph Neural Networks (GNNs) beyond 1-WL expressivity. We describe formally the relation between IGEL and 1-WL, and characterize its expressive power and limitations. Experiments show that IGEL matches the empirical expressivity of state-of-the-art methods on isomorphism detection while improving performance on seven GNN architectures.

## 1 Introduction

Novel approaches for representation learning on graph structured data have appeared in recent years [1]. Graph neural networks can efficiently learn representations that depend both on the graph structure and node and edge features from large-scale graph datasets. The most popular choice of architecture is the Message Passing Graph Neural Network (MP-GNN). In MP-GNNs, a node is represented by iteratively aggregating local feature 'messages' from its neighbors. Despite being succesfully applied in a wide variety of domains [2–6], there is a limit on the representational power of MP-GNNs provided by the computationally efficient Weisfeiler-Lehman (1-WL) test for checking graph isomorphism [7, 8]. Establishing this connection has lead to a better theoretical understanding of the performance of MP-GNNs and many possible generalizations [9–13].

To improve the expressivity of MP-GNNs, recent methods have extended the vanilla message-passing mechanism is various ways. For example, using higher order $k$-vertex tuples [8] leading to $k$-WL generalizations, introducing relative positioning information for network vertices [14], propagating messages beyond direct neighborhoods [15], using concepts from algebraic topology [16], or combining sub-graph information in different ways [17–25]. All aforementioned approaches (which we review in more detail in Appendix A) improve expressivity by extending MP-GNNs architectures, often evaluating on standarized benchmarks [26–29]. However, identifying the optimal approach on novel domains remains unclear and requires costly architecture search.

In this work, we show that incorporating simple ego-network encodings already boosts the expressive power of MP-GNNs beyond the 1-WL test, while keeping the benefits of efficiency and simplicity. We present **IGEL**, an **I**nductive **G**raph **E**ncoding of **L**ocal information, which in its basic form extends node attributes with histograms of node degrees at different distances. The **IGEL** encodings can be computed as a pre-processing step irrespective of model architecture. Theoretically, we formally prove that the **IGEL** encoding is no less expressive than the 1-WL test, and provide examples that show that it is more expressive than 1-WL. We also identify expressivity upper-bounds for graphs that are indistinguishable using state of the art methods. Experimentally, we asses the performance of seven model architectures enriched with **IGEL** encodings on five tasks and ten graph data sets, and find that it consistently improves downstream model performance.

---

[*]Corresponding Author.

N. Alvarez-Gonzalez et al., Beyond 1-WL with Local Ego-Network Encodings (Extended Abstract). Presented at the First Learning on Graphs Conference (LoG 2022), Virtual Event, December 9–12, 2022.

## 2 IGEL: Ego-Networks As Sparse Inductive Representations

Given a graph $G = (V, E)$, we define $n = |V|$ and $m = |E|$, $d_G(v)$ is the degree of a node $v$ in $G$ and $d_{\mathtt{max}}$ is the maximum degree. For $u, v \in V$, $l_G(u, v)$ is their shortest distance, and $\mathtt{diam}(G) = \max(l_G(u, v) | u, v \in V)$ is the diameter of $G$. Double brackets $\{\!\{\cdot\}\!\}$ denote a lexicographically-ordered multi-set, $\mathcal{E}_v^\alpha \subseteq G$ is the $\alpha$-depth ego-network centered on $v$, and $\mathcal{N}_G^\alpha(v)$ is the set of neighbors of $v$ in $G$ up to distance $\alpha$, i.e., $\mathcal{N}_G^\alpha(v) = \{u \mid u \in V \wedge l_G(u, v) \le \alpha\}$.

Algorithm 1 shows the 1-WL test, where $\mathtt{hash}$ maps a multi-set to an equivalence class shared by all nodes with matching multi-set encodings after a 1-WL iteration. The output of 1-WL is $\mathbb{N}^n$—mapping each node to a color, bounded by $n$ distinct colors if each node is uniquely colored. $k$-higher order variants of the WL test (denoted $k$-WL) operate on $k$-tuples of vertices, such that colors are assigned to $k$-vertex tuples. If two graphs $G_1, G_2$ are not distinguishable by the $k$-WL test (that is, their coloring histograms match), they are $k$-WL equivalent—denoted $G_1 \equiv_{k-\mathrm{WL}} G_2$. Due to the hashing step, 1-WL does not preserve distance information in the encoding, and minor changes in the structure of the network (removing one edge) may dramatically change node-level representations. IGEL addresses both limitations, improving expressivity in the process.

### 2.1 The IGEL Algorithm

Intuitively, IGEL encodes a vertex $v$ with the multi-set of ordered degree sequences at each distance within $\mathcal{E}_v^\alpha$. As such, IGEL is a variant of the 1-WL algorithm shown in Algorithm 1, executed for $\alpha$ steps with two modifications. First, the hashing step is removed and replaced by computing the union of multi-sets across steps ($\cup$); second, the iteration number is explicitly introduced in the representation—with the output multi-set $e_v^\alpha$ shown in Algorithm 2.

To be used as vertex features, the multi-set can be represented as a sparse vector $\mathrm{IGEL}_{\mathtt{vec}}^\alpha(v)$, where the frequency of a pair of distance $\lambda$ and degree $\delta$ is contained on index $i = \lambda \cdot (d_{\mathtt{max}} + 1) + \delta$. Degrees greater than $d_{\mathtt{max}}$ are capped to $d_{\mathtt{max}}$, with the resulting vector shown in Figure 1:

$$\mathrm{IGEL}_{\mathtt{vec}}^\alpha(v)_i = \left| \{\!\{(\lambda, \delta) \in e_v^\alpha\}\!\} \right|,$$
$$\text{for } \lambda \cdot (d_{\mathtt{max}} + 1) + \delta = i.$$

$G_1 = (V_1, E_1)$ and $G_2 = (V_1, E_1)$ are IGEL-equivalent for $\alpha$ if the sorted multi-set containing node representations is the same for $G_1$ and $G_2$:

$$G_1 \equiv_{\mathrm{IGEL}}^\alpha G_2 \iff$$
$$\{\!\{e_{v_1}^\alpha : \forall v_1 \in V_1\}\!\} = \{\!\{e_{v_2}^\alpha : \forall v_2 \in V_2\}\!\}.$$

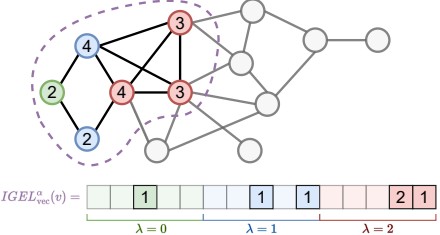

$IGEL_{vec}^\alpha(v) =$

$\lambda = 0 \quad \lambda = 1 \quad \lambda = 2$

**Figure 1:** IGEL encoding of the green vertex. Dashed region denotes $\mathcal{E}_v^\alpha (\alpha = 2)$. The green vertex is at distance $0$, blue vertices at $1$ and red vertices at $2$. Labels show degrees in $\mathcal{E}_v^\alpha$. The frequency of $(\lambda, \delta)$ tuples forming $\mathrm{IGEL}_{\mathtt{vec}}^\alpha(v)$ is: $\{(0, 2) : 1, (1, 2) : 1, (1, 4) : 1, (2, 3) : 2, (2, 4) : 1\}$.

---

**Algorithm 1** 1-WL (Color refinement).

**Input:** $G = (V, E)$
1: $c_v^0 := \mathtt{hash}(\{\!\{d_G(v)\}\!\}) \ \forall \ v \in V$
2: **do**
3: $\quad c_v^{i+1} := \mathtt{hash}(\{\!\{c_u^i : \bigvee\limits_{u \neq v} u \in \mathcal{N}_G^1(v)\}\!\})$
4: **while** $c_v^i \neq c_v^{i-1}$
**Output:** $c_v^i : V \to \mathbb{N}$

---

**Algorithm 2** IGEL Encoding.

**Input:** $G = (V, E), \alpha : \mathbb{N}$
1: $e_v^0 := \{\!\{(0, d_G(v))\}\!\} \ \forall \ v \in V$
2: **for** $i := 1; \ i \mathrel{+}= 1$ **until** $i = \alpha$ **do**
3: $\quad e_v^i := \bigcup(e_v^{i-1},$
4: $\quad\quad \{\!\{(i, d_{\mathcal{E}_G^\alpha(v)}(u))$
5: $\quad\quad \forall u \in \mathcal{N}_G^\alpha(v) \mid l_G(u, v) = i\}\!\})$
6: **end for**
**Output:** $e_v^\alpha : V \to \{\!\{(\mathbb{N}, \mathbb{N})\}\!\}$

---

**Space complexity.** IGEL's worst case space complexity is $\mathcal{O}(\alpha \cdot n \cdot d_{\mathtt{max}})$, conservatively assuming that every node will require $d_{\mathtt{max}}$ parameters at every $\alpha$ depth from the center of the ego-network.

**Time complexity.** For IGEL, each vertex has $d_{\mathtt{max}}$ neighbors where the $\alpha$ iterations imply traversing through geometrically larger ego-networks with $(d_{\mathtt{max}})^\alpha$ vertices, upper bounded by $m$. Thus IGEL's time complexity follows $\mathcal{O}(n \cdot \min(m, (d_{\mathtt{max}})^\alpha))$, with $\mathcal{O}(n \cdot m)$ when $\alpha \ge \mathtt{diam}(G)$, when implemented as BFS, for which we provide further details in Appendix F.

# 3 Theoretical and Experimental Findings

First, we analyze IGEL's expressive power with respect to 1-WL and recent improvements. Second, we measure the impact of IGEL as an additional input to enrich existing MP-GNN architectures.

## 3.1 Expressivity: Which Graphs are IGEL-Distinguishable?

In this section, we discuss the increased expressivity of IGEL with respect to 1-WL, and identify expressivity upper-bounds for graphs that are indistinguishable under MATLANG and the 3-WL test.

**— Relationship to 1-WL.** IGEL is more powerful than 1-WL following Lemma 1 (as described and formally shown in Appendix C) and Lemma 2 (as shown below):

**Lemma 1.** IGEL *is at least as expressive as 1-WL:* $G_1 \not\equiv_{\text{1-WL}} G_2 \Rightarrow G_1 \not\equiv^{\alpha}_{\text{IGEL}} G_2$ *and* $G_1 \equiv^{\alpha}_{\text{IGEL}} G_2 \Rightarrow G_1 \equiv_{\text{1-WL}} G_2$.

**Lemma 2.** *There exist graphs that* IGEL *can distinguish but that 1-WL cannot distinguish.*

*Proof.* For example, any two $d$-regular graphs with equal cardinality are indistinguishable by 1-WL (as shown in Appendix B), but IGEL can distinguish some of them. A graph is $d$-regular if all nodes have degree $d$. Figure 2 shows two $d$-regular graphs where 1-WL (Algorithm 1) assigns the same color for all nodes, stabilizing after one iteration. In contrast, IGEL($\alpha = 1$) counts different frequencies for four structures, hence distinguishing between both graphs. □

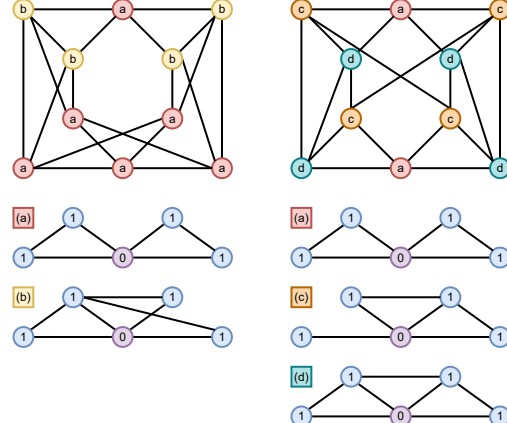

**Figure 2:** IGEL encodings for two Cospectral 4-regular graphs from [30]. IGEL distinguishes 4 kinds of structures within the graphs (associated with every node as a, b, c, and d). The two graphs can be distinguished since the encoded structures and their frequencies do not match.

**— Expressivity upper bounds.** We identify an expressivity upper bound for IGEL, which fails to distinguish **S**trongly **R**egular **G**raphs with equal parameters (Theorem 1, see Appendix E for details):

**Definition 1.** *A $n$-vertex $d$-regular graph is strongly regular—denoted $\text{SRG}(n, d, \beta, \gamma)$—if adjacent vertices have $\beta$ vertices in common, and non-adjacent vertices have $\gamma$ vertices in common.*

**Theorem 1.** IGEL *cannot distinguish $\text{SRG}$s when $n$, $d$, and $\beta$ are the same, and between any value of $\gamma$ (same or otherwise).* IGEL *when $\alpha = 1$ can only distinguish $\text{SRG}$s with different values of $n$, $d$, and $\beta$, while* IGEL *when $\alpha = 2$ can only distinguish $\text{SRG}$s with different values of $n$ and $d$.*

Our findings show that IGEL is a powerful permutation-equivariant representation (see Lemma 3), capable of distinguishing 1-WL equivalent graphs as shown in Figure 2—which as cospectral graphs, are known to be distinguishable in strictly more powerful MATLANG sub-languages than 1-WL [12]. Additionally, the upper bound on $\text{SRG}$s is a hard ceiling on expressivity since $\text{SRG}$s are known to be indistinguishable by 3-WL [31]. IGEL shares the experimental upper-bound of expressivity of methods like GNNML3 [20]. Furthermore, IGEL can provably reach comparable expressivity on $\text{SRG}$s with respect to sub-graph methods implemented within MP-GNN architectures (see Appendix E, subsection E.1), such as Nested GNNs [19] and GNN-AK [23], which are known to be not less powerful than 3-WL, and ESAN when using ego-networks with root-node flags as subgraph sampling policy (EGO+) [24], which is as powerful as the 3-WL test on $\text{SRG}$s (see [24], Prop. 3).

## 3.2 Experimental Evaluation

We evaluate $\text{IGEL}^{\alpha}_{\text{vec}}(v)$ to produce architecture-agnostic vertex features on five tasks: graph classification, isomorphism detection, graphlet counting, link prediction, and node classification.

**Experimental Setup.** We introduce IGEL on graph classification, isomorphism and graphlet counting, comparing the performance of adding/removing IGEL on six GNN architectures following [20]. We also evaluate IGEL on link prediction against transductive baselines, and on node classification as additional feature in MLPs without message-passing. Appendix G describes experimentation details.

**Notation.** The following formatting denotes significant (as per paired t-tests) **positive**, *negative*, and insignificant differences after introducing IGEL, with the best results per task / dataset underlined.

**Table 1:** Per-model graph classification accuracy metrics on TU data sets. Each cell shows the average accuracy of the model and data set in that row and column, with IGEL (left) and without IGEL (right).

| Model | Enzymes | Mutag | Proteins | PTC |
|---|---|---|---|---|
| MLP | **41.10>26.18**$^\diamond$ | **87.61>84.61**$^\diamond$ | 75.43~75.01 | **64.59>62.79**$^\diamond$ |
| GCN | **54.48>48.60**$^\diamond$ | **89.61>85.42**$^\diamond$ | **75.67>74.50**$^*$ | 65.76~65.21 |
| GAT | 54.88~54.95 | **90.00>86.14**$^\diamond$ | **73.44>70.51**$^\diamond$ | 66.29~66.29 |
| GIN | **54.77>53.44**$^*$ | 89.56~88.33 | **73.32>72.05**$^\diamond$ | 61.44~60.21 |
| Chebnet | 61.88~62.23 | **91.44>88.33**$^\diamond$ | **74.30>66.94**$^\diamond$ | 64.79~63.87 |
| GNNML3 | *61.42<62.79*$^\diamond$ | **92.50>91.47**$^*$ | **75.54>62.32**$^\diamond$ | *64.26<66.10*$^\diamond$ |

$*$  $p < 0.01$,  $\diamond$  $p < 0.0001$

**Table 2:** Mean $\pm$ stddev of best IGEL configuration and state-of-the-art results reported on [15, 18, 19, 21, 23, 24] with best performing baselines underlined.

| Model | Mutag | Proteins | PTC |
|---|---|---|---|
| IGEL (ours) | $92.5 \pm 1.2$ | $75.7 \pm 0.3$ | $66.3 \pm 1.3$ |
| *k*-hop [15][†] | $87.9 \pm 1.2^\diamond$ | $75.3 \pm 0.4$ | — |
| GSN [18][†] | $92.2 \pm 7.5$ | $76.6 \pm 5.0$ | $68.2 \pm 7.2$ |
| NGNN [19][†] | $87.9 \pm 8.2$ | $74.2 \pm 3.7$ | — |
| ID-GNN [21][†] | $93.0 \pm 5.6$ | $77.9 \pm 2.4^*$ | $62.5 \pm 5.3$ |
| GNN-AK [23][†] | $91.7 \pm 7.0$ | $77.1 \pm 5.7$ | $67.7 \pm 8.8$ |
| ESAN [24][†] | $91.1 \pm 7.0$ | $76.7 \pm 4.1$ | $69.2 \pm 6.5$ |

†: Results as reported by [15, 18, 19, 21, 23, 24].

**— Graph Classification.** Table 1 shows graph classification results on the TU molecule data sets [27]. We evaluate differences in mean accuracy between 10 runs with (left) / without (right) IGEL. We do not tune network hyper-parameters and establish statistical significance through paired t-tests, with $p < 0.01$ (*) and $p < 0.0001$ ($\diamond$). Our results show that IGEL in the Mutag and Proteins data sets improves the performance of all MP-GNN models, including GNNML3. On the Enzymes and PTC data sets, results are mixed: excluding GNNML3, IGEL either significantly improves accuracy (on MLPNet, GCN, and GIN on Enzymes), or does not have a negative impact on performance.

Table 2 compares IGEL results from Table 1 with reported results for state-of-the-art 1-WL expressive MP-GNNs. Results are comparable to IGEL except where highlighted in color. Overall, when comparing IGEL and best performing baselines, only differences with ID-GNN on Proteins are statistically significant (using $p$-value threshold $p < 0.01$, where ID-GNN shows $p = 0.009$).

**— Isomorphism Detection & Graphlet Counting.** Adding IGEL to the six models in Table 1 on the EXP [32] isomorphism detection yields significant improvements: all GNN models distinguish all non-isomorphic yet 1-WL equivalent EXP graph pairs with IGEL vs. 50% accuracy without IGEL (i.e. random guessing). Additionally, IGEL significantly improves GNN graphlet-counting performance on three graphlet types in the RandomGraph data set [33]. We provide further details in Appendix H.

**— Link Prediction & Node Classification.** We test IGEL on edge / node level tasks to assess its use as a baseline in non-GNN settings. On a transductive link prediction task, we train DeepWalk [34] style embeddings of IGEL encodings rather than node identities on the Facebook and CA-AstroPh graphs [35]. IGEL-derived embeddings outperform transductive baselines on link prediction as an edge-level binary classification task, measuring **0.976** vs. 0.968 (Facebook) and **0.984** vs. 0.937 (CA-AstroPh) AUC comparing IGEL vs. node2vec [36]. On multi-label node classification on PPI [37], we train an MLP (e.g. no message passing) with node features and IGEL encodings. Our MLP shows better micro-F1 (0.850) when $\alpha = 1$ than MP-GNN architectures such as GraphSAGE (0.768, as reported in [38]), but underperforms compared to a 3-layer GAT (**0.973** micro-F1 from [38]).

**— Experimental Summary.** Introducing IGEL yields comparable performance to state-of-the-art methods without architectural modifications—including when compared to strong baseline models focused on WL expressivity such as GNNML3, GSN, Nested GNNs, ID-GNN, GNN-AK or ESAN. Furthermore, IGEL achieves this at a lower computational cost, in comparison for instance with GNNML3, which requires a $\mathcal{O}(n^3)$ eigen-decomposition step to introduce spectral channels. Finally, IGEL can also be used in transductive settings (link prediction) as well as node-level tasks (node classification) and outperform strong transductive baselines or enhance models without message-passing, such as MLPs. As such, we believe IGEL is an attractive baseline with a clear relationship to the 1-WL test that improves MP-GNN expressivity without the need for costly architecture search.

## 4   Conclusions

We presented IGEL, a novel vertex representation algorithm on unattributed graphs allowing MP-GNN architectures to go beyond 1-WL expressivity. We showed that IGEL is related and more expressive than the 1-WL test, and formally proved an expressivity upper bound on certain families of Strongly Regular Graphs. Finally, our experimental results indicate that introducing IGEL in existing MP-GNN architectures yield comparable performance to state-of-the-art methods, without architectural modifications and at lower computational costs than other approaches.

## Author Contributions

Nurudin Alvarez-Gonzalez: Conceptualization, Methodology, Software, Investigation, Formal analysis, Writing - Original Draft; Andreas Kaltenbrunner: Validation, Supervision, Writing - Review & Editing; Vicenç Gómez: Resources, Validation, Supervision, Writing - Review & Editing.

## Acknowledgements

Vicenç Gómez has received funding from "la Caixa" Foundation (ID 100010434), under the agreement LCF/PR/PR16/51110009. Andreas Kaltenbrunner acknowledges support from Intesa Sanpaolo Innovation Center. The funder had no role in study design, data collection and analysis, decision to publish, or preparation of the manuscript.

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

# A    Relation with Previous Works

In the past few years, many different approaches have been developed for improving the expressivity of MP-GNNs. Here we review the works that are more related to IGEL. For a more detailed overview augmented message-passing methods for graph representation learning, see [39].

In $k$**-hop MP-GNNs ($k$-hop)** [15] the authors propose to propagate messages beyond immediate vertex neighbors, effectively using ego-network information in the vertex representation. Their proposed algorithm requires to extract neighborhood sub-graphs and to perform message-passing on each sub-graph, which has an exponential cost on the number of hops $k$ both at pre-processing and at each iteration (epoch). In contrast, IGEL only requires a single pre-processing step that can cached once computed.

Distance Encoding GNNs (**DE-GNN**) [25] also propose to improve MP-GNN by using extra node features by encoding distances to a subset of $p$ nodes. The features obtained by DE-GNN are similar to IGEL when conditioning the subset to size $p = 1$ and using a distance encoding function with $k = \alpha$. However, these features are not strictly equivalent to the IGEL features, since within the ego-network the node degrees can be smaller than the actual degrees, and they are more expensive to compute. DE-GNN needs to compute power iterations of the entire adjacency matrix, which is more expensive and does not exploit network sparsity.

Graph Substructure Networks (**GSNs**) [18] incorporate hand-crafted topological features by counting local substructures (such as the presence of cliques or cycles). GSNs require expert knowledge on what features are relevant for a given task and depart from the original MP-GNN in their architecture. We show that IGEL reaches comparable performance using a general encoding for ego-networks and without altering the original message-passing mechanism.

**GNNML3** [20] proposes a way to perform message passing in spectral-domain with a custom frequency profile. While this approach achieves good performance on graph classification, it requires an expensive preprocessing step for computing the eigendecomposition of the graph Laplacian and $\mathcal{O}(k)$-order tensors to achieve $k$-WL expressiveness, which does not scale to large graphs.

More recently, a series of methods formulate the problem of representing vertices or graphs as aggregations over sub-graphs. The sub-graph information is pooled or introduced during message-passing at an additional cost that varies depending on each architecture. Consequently, they require generating the subgraphs (or effectively replicating the nodes of every subgraph of interest) and pay an additional overhead due to the aggregation. These approaches include **Ego-GNNs** [22], Nested GNNs (**NGNNs**) [19], GNN-as-Kernel (**GNN-AK**) [23], Identity-aware GNNs (**ID-GNNs**) [21].

**Ego-GNNs** perform message-passing over the ego-graphs of all the nodes in a graph, and subsequently perform aggregation. They provide empirical evidence of a superior expressive power than the classical 1-WL. **ID-GNNs** embed each node incorporating identity information in the GNN and apply rounds of heterogeneous message passing; **NGNNs** perform a two-level GNN using rooted sub-graphs and consider a graph as a bag of sub-graphs; **GNN-AK** uses a very similar idea, but as the authors describe, it sets the number of layers to the number of iterations of 1-WL; Compared to all these methods IGEL only relies on an initial pre-processing step based on distances and degrees without having to run additional message passing iterations. Despite its simplicity, IGEL performs competitively, as we show in Table 2.

Equivariant Subgraph Aggregation Networks (**ESAN**) [24] also propose to encode bags of subgraphs and show that such an encoding can lead to a better expressive power. In the case of the ego-networks policy (EGO), **ESAN** is strongly related with IGEL. Interestingly, as described in concurrent work [40], the implicit encoding of the pairwise distance between nodes, plus the degree information which can be extracted via aggregation are fundamental to provide a theoretical justification of **ESAN**. In this work, we directly consider distances and degrees in the ego-network, explicitly providing the structural information encoded by more expressive GNN architectures. These similarities may explain why the performance of both methods is comparable, as shown in Table 2.

# B    1-WL Expressivity and Regular Graphs

Remark 1 shows that 1-WL, as defined in Algorithm 1, is unable of distinguishing $d$-regular graphs:

**Remark 1.** *Let $G_1$ and $G_2$ be two $d$-regular graphs such that $|V_1| = |V_2|$. Tracing Algorithm 1, all vertices in $V_1$, $V_2$ share the same initial color due to $d$-regularity: $\forall v \in V_1 \bigcup V_2; c_v^0 = \mathtt{hash}(\{\{d\}\})$.*

*After the first color refinement iteration, consider the colorings of $G_1$ and $G_2$:*

— $\forall\, v_1 \in V_1; c_{v_1}^1 := \mathtt{hash}(\{\!\{c_{u_1}^0 : \underset{u_1 \neq v_1}{\forall}\; u_1 \in \mathcal{N}_{G_1}^1(v_1)\}\!\}),$

— $\forall\, v_2 \in V_2; c_{v_2}^1 := \mathtt{hash}(\{\!\{c_{u_2}^0 : \underset{u_2 \neq v_2}{\forall}\; u_2 \in \mathcal{N}_{G_2}^1(v_2)\}\!\}).$

*Since $\forall\, v_1 \in V_1, v_2 \in V_2; d = |\mathcal{N}_{G_1}^1(v_1)| = |\mathcal{N}_{G_2}^1(v_2)|$, substituting $c_{v_1}^1$, $c_{v_2}^1$ in the next iteration step yields $\{\!\{\mathtt{hash}(c_{v_1}^1) : \forall\, v_1 \in V_1\}\!\} = \{\!\{\mathtt{hash}(c_{v_2}^1) : \forall\, v_2 \in V_2\}\!\}$. Thus, on any pair of $d$-regular graphs with equal cardinality, 1-WL stabilizes after one iteration produces equal colorings for all nodes on both graphs—regardless of whether $G_1$ and $G_2$ are isomorphic, as Figure 2 shows.*

## C    IGEL is At Least As Powerful as 1-WL

In this section we formally prove Lemma 1, i.e. that IGEL is at least as expressive as 1-WL. For this, we consider a variant of 1-WL which removes the hashing step. This modification can only increase the expressive power of 1-WL but makes it possible to directly compare with the encodings generated by IGEL. Intuitively, after $k$ color refinement iterations, 1-WL considers nodes at $k$ hops from each node, which is equivalent to running IGEL with $\alpha = k + 1$, so that the ego-networks include the information of all nodes that 1-WL would visit.

**Lemma 1.** IGEL *is at least as expressive as 1-WL. For two graphs $G_1$, $G_2$ which are distinguished by 1-WL in $k$ iterations ($G_1 \not\equiv_{1\text{-}WL} G_2$) it also holds that $G_1 \not\equiv_{\text{IGEL}}^\alpha G_2$ for $\alpha = k + 1$. If IGEL does not distinguish two graphs $G_1'$ and $G_2'$, 1-WL also does not distinguish them: $G_1' \equiv_{\text{IGEL}}^\alpha G_2' \Rightarrow G_1' \equiv_{1\text{-}WL} G_2'$.*

*Proof of Lemma 1:* For convenience, let $c_v^{i+1} = \{\!\{c_v^i; c_u^i \;\forall\, u \in \mathcal{N}_G^1(v) \mid u \neq v\}\!\}$ be a recursive definition of Algorithm 1 where hashing is removed and $c_v^0 = \{\!\{d_G(v)\}\!\}$. Since the hash is no longer computed, the nested multi-sets contain strictly the same or more information as in the traditional 1-WL algorithm.

For IGEL to be less expressive than 1-WL, it must hold that there exist two graphs $G_1 = (V_1, E_1)$ and $G_2 = (V_2, E_2)$ such that $G_1 \not\equiv_{1\text{-}WL} G_2$ while $G_1 \equiv_{\text{IGEL}}^\alpha G_2$.

Let $k$ be the minimum number of color refinement iterations such that $\exists\, v_1 \in V_1$ and $\forall\, v_2 \in V_2, c_{v_1}^k \neq c_{v_2}^k$. We define an equally or more expressive variant of the 1-WL test 1-WL$^*$ where hashing is removed, such that $c_{v_1}^k = \{\!\{\{\!\{...\{\!\{d_G(v_1)\}\!\}, \{\!\{d_G(u)\forall u \in \mathcal{N}_{G_1}^1(v_1)\}\!\}...\}\!\}\}\!\}$, nested up to depth $k$. To avoid nesting, the multi-set of nested degree multi-sets can be rewritten as the union of degree multi-sets by introducing an indicator variable for the iteration number where a degree is found:

$$
c_{v_1}^k = \left\{\!\!\left\{ (0, d_G(v_1)) \right\}\!\!\right\} \bigcup
$$
$$
\left\{\!\!\left\{ (1, d_G(v_1)); (1, d_G(u)) \,\forall\, u \in \mathcal{N}_G^1(v_1) \right\}\!\!\right\} \bigcup
$$
$$
\left\{\!\!\left\{ (2, d_G(v_1)); (2, d_G(u)) \,\forall\, u \in \mathcal{N}_G^1(v_1); (2, d_G(w)) \,\forall\, w \in \mathcal{N}_G^1(u) \right\}\!\!\right\} \bigcup ...
$$

At each step $i$, we introduce information about nodes up to distance $i$ of $v_1$. Furthermore, by construction, nodes will be visited on every subsequent iteration—i.e. for $c_{v_1}^2$, we will observe $(2, d_G(v_1))$ exactly $d_G(v_1) + 1$ times, as all its $d_G(v_1)$ neighbors $u \in \mathcal{N}_G^1(v)$ encode the degree of $v_1$ in $c_u^1$. The flattened representation provided by 1-WL$^*$ is still equally or more expressive than 1-WL, as it removes hashing and keeps track of the iteration at which a degree is found.

Let IGEL-W be a less expressive version of IGEL that does not include edges between nodes at $k + 1$ hops of the ego-network center. Now consider the case in which $c_{v_1}^k \neq c_{v_2}^k$ from 1-WL$^*$, and let $\alpha = k + 1$ so that IGEL-W considers degrees by counting edges found at $k$ to $k + 1$ hops of $v_1$ and $v_2$. Assume that $G_1 \equiv_{\text{IGEL-W}}^\alpha G_2$. By construction, this means that $\{\!\{e_{v_1}^\alpha : \forall\, v_1 \in V_1\}\!\} = \{\!\{e_{v_2}^\alpha : \forall\, v_2 \in V_2\}\!\}$. This implies that all degrees and iteration counts match as per the distance indicator variable at which the degrees are found, so $c_{v_1}^k = c_{v_2}^k$ which contradicts the assumption $c_{v_1}^k \neq c_{v_2}^k$ and therefore implies that also $G_1 \equiv_{1\text{-}WL^*} G_2$. Thus, $G_1 \equiv_{\text{IGEL-W}}^\alpha G_2 \Rightarrow G_1 \equiv_{1\text{-}WL^*} G_2$ for $\alpha = k + 1$ and also $G_1 \not\equiv_{1\text{-}WL^*} G_2 \Rightarrow G_1 \not\equiv_{\text{IGEL-W}}^\alpha G_2$. Therefore by extension IGEL is at least as expressive as 1-WL. $\qquad\square$

# D  IGEL is Permutation Equivariant

**Lemma 3.** *Given any $v \in V$ for $G = (V, E)$ and given a permuted graph $G' = (V', E')$ of $G$ produced by a permutation of node labels $\pi : V \to V'$ such that $\forall v \in V \Leftrightarrow \pi(v) \in V'$, $\forall (u, v) \in E \Leftrightarrow (\pi(u), \pi(v)) \in E'$.*

*The IGEL representation is permutation equivariant at the graph level*

$$\pi(\{\!\{e_{v_1}^{\alpha}, \ldots, e_{v_n}^{\alpha}\}\!\}) = \{\!\{e_{\pi(v_1)}^{\alpha}, \ldots, e_{\pi(v_n)}^{\alpha}\}\!\}.$$

*The IGEL representation is permutation invariant at the node level*

$$e_v^{\alpha} = e_{\pi(v)}^{\alpha}, \forall v \in G.$$

*Proof.* Note that $e_v^{\alpha}$ in Algorithm 2 can be expressed recursively as:

$$e_v^{\alpha} = \left\{\!\!\left\{ \left( l_{\mathcal{E}_v^{\alpha}}(u, v), d_{\mathcal{E}_v^{\alpha}}(u) \right) \middle| \forall u \in \mathcal{N}_G^{\alpha}(v) \right\}\!\!\right\}.$$

Since IGEL only relies on node distances $l_G(\cdot, \cdot)$ and degree nodes $d_G(\cdot)$, and both $l_G(\cdot, \cdot)$ and $d_G(\cdot)$ are permutation invariant (and the node level) and equivariant (at the graph level) functions, the IGEL representation is permutation equivariant at the graph level, and permutation invariant at the node level. $\square$

# E  Proof of Theorem 1

In this appendix, we provide proof for Theorem 1, showing that IGEL cannot distinguish certain pairs of SRGs with equal parameters of $n$ (cardinality), $d$ (degree), $\beta$ (shared edges between adjacent nodes), and $\gamma$ (shared edges between non-adjacent nodes). Let $\{\!\{\cdot\}\!\}^d$ denote a repeated multi-set with $d$-times the cardinality of the items in the multi-set, and let $e_G^{\alpha} = \{\!\{e_v^{\alpha} : \forall v \in V\}\!\}$ be short-hand notation for the IGEL encoding of $G$, defined as the sorted multi-set containing IGEL encodings of all nodes in $G$.

**Lemma 4.** *For any $G = SRG(n, d, \beta, \gamma)$, $diam(G) \leq 2$.*

*Note that by definition of SRGs, $n$ affects cardinality while $d$ and $\beta$ control adjacent vertex connectivity at 1-hop. For $\gamma$, we have to consider two cases: when $\gamma \geq 1$ and when $\gamma = 0$:*

*— Let $\gamma \geq 1$: by definition, $\forall u, v \in V s.t. (u, v) \notin E, \exists w \in V s.t. (u, w) \in E \land (v, w) \in E$. Thus, $\forall (u, v) \in E, l_G(u, v) = 1$ and $\forall (u, v) \notin E, l_G(u, v) = 2$.*

*— Let $\gamma = 0$: $\forall u, v \in V$, if $(u, v) \notin E$ then $\nexists w \in V s.t. (u, w) \in E \land (v, w) \in E$ as $w$ is in common between $u$ and $v$. Then, $\forall u, v, w \in V s.t. (u, v) \in E, (u, w) \in E \Leftrightarrow (v, w) \in E$—hence, only nodes and their neighbors can be in common. Thus: $\forall u, v \in V s.t. u \neq v, l_G(u, v) = 1$.*

*Given both scenarios, we can conclude that for any $\gamma \in \mathbb{N}$, $\forall u, v \in V, l_G(u, v) \leq 2$ and thus $diam(G) \leq 2$.* $\square$

**Lemma 5.** *For any finite graph $G$, there is a finite range of $\alpha \in \mathbb{N}$ where IGEL encodings distinguish between different values of $\alpha$. For values of $\alpha$ larger than the diameter of the graph (that is, $\alpha \geq diam(G)$), it holds that $e_v^{\alpha} = e_v^{\alpha+1}$ as $\mathcal{E}_v^{\alpha} = \mathcal{E}_v^{\alpha+1} = G$.* $\square$

*Proof.* Per Lemma 4 and Lemma 5, SRGs have a maximum diameter of two, and IGEL encodings are equal for all $\alpha \geq \text{diam}(G)$. Thus, given $G = SRG(n, d, \beta, \gamma)$, only $\alpha \in \{1, 2\}$ produce different encodings of $G$. It can be shown that $e_v^{\alpha}$ can only distinguish different values of $n$, $d$ and $\beta$, and $IGEL_{enc}^2$ can only distinguish values of $n$ and $d$:

*— Let $\alpha = 1$: $\forall v \in V, \mathcal{E}_v^1 = (V', E')$ s.t. $V' = \mathcal{N}_G^1(v)$. Since $G$ is $d$-regular, $v$ is the center of $\mathcal{E}_v^1$, and has $d$-neighbors. By SRG's definition, the $d$ neighbors of $v$ have $\beta$ shared neighbors with $v$ each, plus an edge with $v$. Thus, for any SRGs $G_1, G_2$ where $n_1 = n_2$, $d_1 = d_2$, and $\beta_1 = \beta_2$, $e_{G_1}^1 = e_{G_2}^1$ produce equal encodings by expanding $e_v^1$ in Algorithm 2:

$$e_v^1 = \left\{\!\!\left\{ (0, d) \right\}\!\!\right\} \bigcup \left\{\!\!\left\{ (1, \beta + 1) \right\}\!\!\right\}^d$$

— Let $\alpha = 2$: $\forall v \in V, \mathcal{E}_v^2 = G$ as $\forall u \in V, u \in \mathcal{N}_G^2(v)$ when $\texttt{diam}(G) \leq 2$. $G$ is $d$-regular, so $\forall v \in V, d = d_{\mathcal{E}_v^2}(v) = d_G(v)$. Thus, for any SRGs $G_1, G_2$ s.t. $n_1 = n_2$ and $d_1 = d_2$, $e_{G_1}^2 = e_{G_1}^2$, containing $n$ equal $e_v^2$ encodings by expanding Algorithm 2:

$$e_v^2 = \left\{\!\!\left\{ (0, d) \right\}\!\!\right\} \bigcup \left\{\!\!\left\{ (1, d) \right\}\!\!\right\}^d \bigcup \left\{\!\!\left\{ (2, d) \right\}\!\!\right\}^{n-d-1}$$

Thus, IGEL cannot distinguish pairs of SRGs when $n$, $d$, and $\beta$ are the same, and between any value of $\gamma$ (equal or different between the pair). IGEL when $\alpha = 1$ can only distinguish SRGs with different values of $n$, $d$, and $\beta$, while IGEL when $\alpha = 2$ can only distinguish SRGs with different values of $n$ and $d$. $\qquad\square$

We note that it is straightforward to extend IGEL so that different values of $\gamma$ can be distinguished. We explore one possible extension in subsection E.1.

### E.1  Improving Expressivity on the $\gamma$ Parameter

IGEL as presented is unable to distinguish between any values of $\gamma$ in SRGs. However, IGEL can be trivially extended to distinguish between pairs of SRGs, bringing parity with methods such as the EGO+ policy in ESAN, NGNNs and GNN-AK.

Intuitively, IGEL is unable to distinguish $\gamma$ because its $(\lambda, \delta)$ tuples are unable to represent relationships between vertices at different distances (i.e. the $\gamma$ parameter). The structural feature definition may be extended to compute the degree between 'distance layers' in the sub-graphs, addressing this pitfall. This means modifying $e_v^i$ in Algorithm 2:

$$e_v^i = e_v^{i-1} \cup \left\{\!\!\left\{ \rho(u, v) : \forall u \in \mathcal{N}_G^\alpha(v) \,\Big|\, l_G(u, v) \in \{i, i+1\} \right\}\!\!\right\}$$

where:

$$\rho(u, v) = \left( l_{\mathcal{E}_v^\alpha}(u, v), d_{\mathcal{E}_v^\alpha}^0(u, v), d_{\mathcal{E}_v^\alpha}^1(u, v) \right)$$

and $d_G^p(u, v)$ generalizes $d_G(u)$ to count edges of $u$ at a relative distance $p$ of $v$ in $G = (V, E)$:

$$d_G^p(u, v) = \Big| (u, w) \in E \,\forall\, w \in V \, s.t. \, l_G(u, w) = l_G(u, v) + p \Big|.$$

It can be shown that this definition of $e_v^i$ is strictly more powerful distinguishing at SRGs following an expansion of Algorithm 2 with $\alpha = 2$:

$$e_v^2 = \left\{\!\!\left\{ (0, 0, d) \right\}\!\!\right\} \bigcup \left\{\!\!\left\{ (1, \beta, \gamma) \right\}\!\!\right\}^d \bigcup \left\{\!\!\left\{ (2, d - \gamma, 0) \right\}\!\!\right\}^{n-d-1}$$

*Proof.* For any $G = \texttt{SRG}(n, d, \beta, \gamma)$, $\forall v \in V$, $l_{\mathcal{E}_v^2}(v, v) = 0$ and there are $d$ edges towards its neighbors—thus the root is encoded as $(0, 0, d)$. Each neighbor is at $l_{\mathcal{E}_v^2}(u, v) = 1$, with $\beta$ edges among each other, and $\gamma$ with vertices not adjacent to $v$—thus $(1, \beta, \gamma)$, where $d = 1 + \beta + \gamma$. By definition, every vertex $w \in V \, s.t. (u, w) \notin E$ has $\gamma$ neighbors shared with $v$, and $d$ neighbors overall. Per Lemma 4, the maximum diameter of $G$ is two, hence $l_{\mathcal{E}_v^2}(v, w) = 2$ and for any $w$, the representation is $(2, d - \gamma, 0)$. $\qquad\square$

## F  Implementing IGEL through Breadth-First Search

The idea behind the IGEL encoding is to represent each vertex $v$ by compactly encoding its corresponding ego-network $\mathcal{E}_v^\alpha$ at depth $\alpha$. The choice of encoding consists of a histogram of vertex degrees at distance $d \leq \alpha$, for each vertex in $\mathcal{E}_v^\alpha$. Essentially, IGEL runs a Breadth-First Traversal up to depth $\alpha$, counting the number of times the same degree appears at distance $d \leq \alpha$.

The algorithm shown in Algorithm 2 showcases IGEL and its relationship to the 1-WL test. However, in a practical setting, it might be preferable to implement IGEL through Breadth-First Search (BFS). In Algorithm 3, we show one such implementation that fits the time and space complexity described in section 2:

---

**Algorithm 3** IGEL Encoding (BFS).

---

**Input:** $v \in V, \alpha \in \mathbb{N}$
1: toVisit := [ ]                                                                                    ▷ Queue of nodes to visit.
2: degrees := { }                                                                          ▷ Mapping of nodes to their degrees.
3: distances := $\{v : 0\}$                                                    ▷ Mapping of nodes to their distance to $v$
4: **while** toVisit $\neq \emptyset$ **do**
5:     $u :=$ toVisit.dequeue()
6:     currentDistance := distances$[u]$
7:     currentDegree := 0
8:     **for** $w \in u$.neighbors() **do**
9:         **if** $w \notin$ distances **then**
10:             distances$[w] :=$ currentDistance $+ 1$  ▷ $w$ is a new node 1-hop further from $v$.
11:         **end if**
12:         **if** distances$[w] \leq \alpha$ **then**
13:             currentDegree := currentDegree $+ 1$          ▷ Count edges only within $\alpha$-hops.
14:             **if** $w \notin$ degrees **then**                                ▷ Enqueue if $w$ has not been visited.
15:                 toVisit.append$(w)$
16:             **end if**
17:         **end if**
18:     **end for**
19:     degrees$[u] :=$ currentDegree   ▷ $u$ is now visited: we know its degree and distance to $v$.
20: **end while**
21: $e_v^\alpha = \{\!\{($distances$[u],$ degrees$[u]) \, \forall \, u \in$ degrees.keys()$\}\!\}$
                                    ▷ Produce the multi-set of (distance, degree) pairs for all visited nodes.
**Output:** $e_v^\alpha : (\mathbb{N}, \mathbb{N}) \to \mathbb{N}$

---

Due to how we structure BFS to count degrees and distances in a single pass, each edge is processed twice—once for each node at end of the edge. It must be noted that when processing every $v \in V$, the time complexity is $\mathcal{O}(n \cdot \min(m, (d_{\texttt{max}})^\alpha))$. However, the BFS implementation is also embarrassingly parallel, which means that it can be distributed over $p$ processors with $\mathcal{O}(n \cdot \min(m, (d_{\texttt{max}})^\alpha)/p)$ time complexity.

## G   Experimental Settings And Procedures

In this section, we provide additional details of our experimental setting. We summarize our datasets and tasks in Table 6.

On graph-level tasks, we introduce IGEL encodings concatenated to existing vertex features into the best performing model configurations found by [20] without any hyper-parameter tuning (e.g. number of layers, hidden units, choice pooling and activation functions). We evaluate performance differences with and without IGEL on each task, data set and model on 10 independent runs, measuring statistical significance of the differences through paired t-tests.

On vertex and edge-level tasks, we report best performing configurations after hyper-parameter search. Each configuration is evaluated on 5 independent runs. We provide a breakdown of the best performing hyper-parameters in the section below.

### G.1   Hyper-parameters and Experiment Details

**Graph Level Experiments**

We reproduce the benchmark of [20] without modifying model hyper-parameters for the tasks of Graph Classification, Graph Isomorphism Detection, and Graphlet Counting. For classification tasks, the 6 models in Table 2 are trained on binary / categorical cross-entropy objectives depending on the task. For Graph Isomorphism Detection, we train GNNs as binary classification models on the binary classification task on EXP [32], and identify isomorphisms by counting the number of graph pairs

**Table 3:** Values of $\alpha$ used when introducing IGEL in the best reported configuration for graphlet counting and graph classification tasks. The table is broken down by graphlet types (upper section) and graph classification tasks on the TU Datasets (bottom section).

|  | Chebnet | GAT | GCN | GIN | GNNML3 | Linear | MLP |
|---|---|---|---|---|---|---|---|
| **Star** | 2 | 1 | 2 | 1 | 1 | 2 | 1 |
| **Tailed Triangle** | 1 | 1 | 1 | 1 | 2 | 1 | 1 |
| **Triangle** | 1 | 1 | 1 | 1 | 1 | 1 | 1 |
| **4-Cycle** | 2 | 1 | 1 | 1 | 1 | 1 | 1 |
| **Custom Graphlet** | 2 | 1 | 1 | 1 | 2 | 2 | 2 |
| **Enzymes** | 1 | 2 | 2 | 1 | 2 | 2 | 2 |
| **Mutag** | 1 | 1 | 1 | 1 | 1 | 1 | 2 |
| **Proteins** | 2 | 2 | 2 | 1 | 2 | 1 | 1 |
| **PTC** | 1 | 1 | 2 | 1 | 1 | 2 | 2 |

for which randomly initialized MP-GNN models produce equivalent outputs on Graph8c[23]. For the graphlet counting regression task on the RandomGraph data set [33], we train models to minimize Mean Squared Error (MSE) on the normalized graphlet counts[4] for five types of graphlets.

On all tasks, we experiment with $\alpha \in \{1, 2\}$ and optionally introduce a preliminary linear transformation layer to reduce the dimensionality of IGEL encodings. For every setup, we execute the same configuration 10 times with different seeds and compare runs introducing IGEL or not by measuring whether differences on the target metric (e.g. accuracy or MSE) are statistically significant as shown in Table 1 and Table 2. In Table 3, we provide the value of $\alpha$ that was used in our experimental results. Our results show that the choice of $\alpha$ depends on both the task and model type. We believe these results may be applicable to subgraph-based MP-GNNs, and will explore how different settings, graph sizes, and downstream models interact with $\alpha$ in future work.

*Reproducibility–* We provide an additional repository with our changes to the original benchmark, including our modelling scripts, metadata, and experimental results[5].

**Vertex and Edge-level Experiments**

In this section we break down the best performing hyper-parameters on the Edge (link prediction) and Vertex-level (node classification) experiments.

*Link Prediction–* The best performing hyperparameter configuration on the Facebook graph including $\alpha = 2$, learning $t = 256$ component vectors with $e = 10$ walks per node, each of length $s = 150$ and $p = 8$ negative samples per positive for the self-supervised negative sampling. Respectively on the arXiv citation graph, we find the best configuration at $\alpha = 2$, $t = 256$, $e = 2$, $s = 100$ and $p = 9$.

*Node Classification–* We analyze both encoding distances $\alpha \in \{1, 2\}$. Other IGEL hyper-parameters are fixed after a small greedy search based on the best configurations in the link prediction experiments. For the MLP model, we perform greedy architecture search, including number of hidden units, activation functions and depth. Our results show scores averaged over five different seeded runs with the same configuration obtained from hyperparameter search.

The best performing hyperparameter configuration on the node classification is found with $\alpha = 2$ on $t = 256$ length embedding vectors, concatenated with node features as the input layer for 1000 epochs in a 3-layer MLP using ELU activations with a learning rate of 0.005. Additionally, we apply 100 epoch patience for early stopping, monitoring the F1-score on the validation set.

*Reproducibility–* We provide a replication folder in the code repository for the exact configurations used to run the experiments[6].

---

[2]Simple 8 vertices graphs from: http://users.cecs.anu.edu.au/~bdm/data/graphs.html

[3]That is, models are not trained but simply initialized, following the approach of [20].

[4]Counts are normalized by the standard deviation counts across the data set for MSE values to be consistent across graphlet types, in alignment with [20].

[5]https://github.com/nur-ag/gnn-matlang

[6]https://github.com/nur-ag/IGEL

# H   Extended Results on Isomorphism Detection and Graphlet Counting

In this section we summarize additional results on isomorphism detection and graphlet counting.

## H.1   Isomorphism Detection

We provide a detailed breakdown of isomorphism detection performance after introducing IGEL in Table 4, complimenting our summary on subsection 3.2.

— **Graph8c.** On the Graph8c dataset, introducing IGEL significantly reduces the amount of graph pairs erroneously identified as isomorphic for all MP-GNN models, as shown in Table 4. Furthermore, IGEL allows a linear baseline employing a sum readout function over input feature vectors, then projecting onto a 10-component space, to identify all but 1571 non-isomorphic pairs compared to the erroneous pairs GCNs (4196 errors) or GATs (1827 errors) can identify without IGEL. Additionally, we find that all Graph8c graphs can be distinguished if the IGEL encodings for $\alpha = 1$ and $\alpha = 2$ are concatenated. We do not explore the expressivity of combinations of $\alpha$ in this work, but hypothesize that concatenated encodings of $\alpha$ may be more expressive.

— **Empirical Results on Strongly Regular Graphs.** We also evaluate IGEL on SR25[7], which contains 15 Strongly Regular graphs with 25 vertices, known to be indistinguishable by 3-WL. With SR25, we validate Theorem 1. [20] showed that no models in our benchmark distinguish any of the 105 non-isomorphic graph pairs in SR25. As expected from Theorem 1, IGEL does not improve distinguishability.

**Table 4:** Graph isomorphism detection results. The IGEL column denotes whether IGEL is used or not in the configuration. For Graph8c, we describe graph pairs erroneously detected as isomorphic. For EXP classify, we show the accuracy of distinguishing non-isomorphic graphs in a binary classification task.

| Model | + IGEL | Graph8c (#Errors) | EXP Classify (Accuracy) |
|---|---|---|---|
| Linear | No | 6.242M | 50% |
| | Yes | **1571** | **97.25%** |
| MLP | No | 293K | 50% |
| | Yes | **1487** | **100%** |
| GCN | No | 4196 | 50% |
| | Yes | **5** | **100%** |
| GAT | No | 1827 | 50% |
| | Yes | **5** | **100%** |
| GIN | No | 571 | 50% |
| | Yes | **5** | **100%** |
| Chebnet | No | 44 | 50% |
| | Yes | **1** | **100%** |
| GNNML3 | No | 0 | 100% |
| | Yes | 0 | 100% |

## H.2   Graphlet Counting

We evaluate IGEL on a (regression) graphlet[8] counting task. We minimize Mean Squared Error (MSE) on normalized graphlet counts[9]. Table 5 shows the results of introducing IGEL in 5 graphlet counting tasks on the RandomGraph data set [33]. Stat sig. differences ($p < 0.0001$) shown in **bold green**, with best (lowest MSE) per-graphlet results underlined.

Introducing IGEL improves counting performance on triangles, tailed triangles and the custom 1-WL graphlets proposed by [20]. Star graphlets can be identified by all baselines, and IGEL only produces statistically significant improvements for the Linear baseline.

Notably, the Linear baseline plus IGEL outperforms MP-GNNs without IGEL for star, triangle, tailed triangle and custom 1-WL graphlets. By introducing IGEL on the MLP baseline, it outperforms all other models including GNNML3 on the triangle, tailed-triangle and custom 1-WL graphlets.

**Table 5:** Graphlet counting results. Cells contain mean test set MSE error (lower is better), stat. sig **highlighted**.

| Model | + IGEL | Star | Triangle | Tailed Tri. | 4-Cycle | Custom |
|---|---|---|---|---|---|---|
| Linear | No | 1.60E-01 | 3.41E-01 | 2.82E-01 | 2.03E-01 | 5.11E-01 |
| | Yes | **4.23E-03** | **4.38E-03** | **1.85E-02** | 1.36E-01 | **5.25E-02** |
| MLP | No | 2.66E-06 | 2.56E-01 | 1.60E-01 | 1.18E-01 | 4.54E-04 |
| | Yes | 8.31E-05 | **5.69E-05** | **5.57E-05** | 7.64E-02 | **2.34E-04** |
| GCN | No | 4.72E-04 | 2.42E-01 | 1.35E-01 | 1.11E-01 | 1.54E-03 |
| | Yes | 8.26E-04 | **1.25E-03** | **4.15E-03** | 7.32E-02 | **1.17E-03** |
| GAT | No | 4.15E-04 | 2.35E-01 | 1.28E-01 | 1.11E-01 | 2.85E-03 |
| | Yes | 4.52E-04 | **6.22E-04** | **7.77E-04** | 7.33E-02 | **6.66E-04** |
| GIN | No | 3.17E-04 | 2.26E-01 | 1.22E-01 | 1.11E-01 | 2.69E-03 |
| | Yes | 6.09E-04 | **1.03E-03** | **2.72E-03** | 6.98E-02 | **2.18E-03** |
| Chebnet | No | 5.79E-04 | 1.71E-01 | 1.12E-01 | 8.95E-02 | 2.06E-03 |
| | Yes | 3.81E-03 | **7.88E-04** | **2.10E-03** | 7.90E-02 | **2.05E-03** |
| GNNML3 | No | 8.90E-05 | 2.36E-04 | 2.91E-04 | 6.82E-04 | 9.86E-04 |
| | Yes | 9.29E-04 | 2.19E-04 | 4.23E-04 | 6.98E-04 | 4.17E-04 |

---

[7]SRG$(25, 12, 5, 6)$ graphs from: http://users.cecs.anu.edu.au/~bdm/data/graphs.html

[8]3-stars, triangles, tailed triangles and 4-cycles, plus a custom 1-WL graphlet proposed in [20]

[9]Counts are stddev-normalized so that MSE values are comparable across graphlet types, following [20].

Since Linear and MLP baselines do not use message passing, we believe raw IGEL encodings may be sufficient to identify certain graph structures even with simple linear models. For all graphlets except 4-cycles, introducing IGEL yields performance similar to GNNML3 at lower pre-processing and model training/inference costs, as IGEL obviates the need for costly eigen-decomposition and can be used in simple models only performing graph-level readouts without message passing.

**Table 6:** Overview of the graphs used in the experiments. We show the average number of vertices (Avg. $n$), edges (Avg. $m$), number of graphs, target task, output shape, and splits (when applicable).

| | Avg. $n$ | Avg. $m$ | Num. Graphs | Task | Output Shape | Splits (Train / Valid / Test) |
|---|---|---|---|---|---|---|
| **Enzymes** | 32.63 | 62.14 | 600 | Multi-class Graph Class. | 6 (multi-class probabilities) | 9-fold / 1 fold (Graphs, Train / Eval) |
| **Mutag** | 17.93 | 39.58 | 188 | Binary Graph Class. | 2 (binary class probabilities) | 9-fold / 1 fold (Graphs, Train / Eval) |
| **Proteins** | 39.06 | 72.82 | 1113 | Binary Graph Class. | 2 (binary class probabilities) | 9-fold / 1 fold (Graphs, Train / Eval) |
| **PTC** | 25.55 | 51.92 | 344 | Binary Graph Class. | 2 (binary class probabilities) | 9-fold / 1 fold (Graphs, Train / Eval) |
| **Graph8c** | 8.0 | 28.82 | 11117 | Non-isomorphism Detection | N/A | N/A |
| **EXP Classify** | 44.44 | 111.21 | 600 | Binary Class. (pairwise graph distinguishability) | 1 (non-isomorphic graph pair probability) | Graph pairs 400 / 100 / 100 |
| **SR25** | 25 | 300 | 15 | Non-isomorphism Detection | N/A | N/A |
| **RandomGraph** | 18.8 | 62.67 | 5000 | Regression (Graphlet Counting) | 1 (graphlet counts) | Graphs 1500 / 1000 / 2500 |
| **ArXiv ASTRO-PH** | 18722 | 198110 | 1 | Binary Class. (Link Prediction) | 1 (edge probability) | Randomly sampled edges 50% train / 50% test |
| **Facebook** | 4039 | 88234 | 1 | Binary Class. (Link Prediction) | 1 (edge probability) | Randomly sampled edges 50% train / 50% test |
| **PPI** | 2373 | 68342.4 | 24 | Multi-label Vertex Class. | 121 (binary class probabilities) | Graphs 20 / 2 / 2 |

