# OpenReview forum: "Beyond 1-WL with Local Ego-Network Encodings"
_logconference.io/LOG/2022/Conference — LoG 2022 Poster_

### Official Review · Reviewer_gaQH · 2022-10-06

**Overall Score:** 6
**Confidence:** 5

**Review:**

### **Summary**
The authors propose IGEL, a variant of the standard Colour Refinement algorithm (a.k.a. 1-WL) which encodes rooted ego-networks. Effectively, IGEL represents ego-networks by computing the frequency of nodes of a certain degree and at a specific distance from the root. The rooted ego-network representations can be bijectively mapped to the nodes in the vertex set, producing node features which, as the authors show, can be more discriminative that those in output from 1-WL. These features can be used as initial node representations in, potentially, any downstream graph representation learning algorithm — or can be used to augmented original node features, if any. The authors experimentally show that IGEL node representations generally improve the performance of many graph-learning algorithms on several tasks. In addition, the authors theoretically analyse the expressiveness limitations of IGEL by studying their separation ability over families of Strongly Regular Graphs.

### **Recommendation**
The manuscript presents sound results of limited theoretical interest but moderate practical application. The overall paper is well executed and clearly presented. However, I believe it mostly lacks proper positioning and comparison w.r.t. previous related works. Currently, my decision is to reject the paper, but could be persuaded otherwise if the authors address at least the most relevant points highlighted below (see “Weaknesses”, “Questions”, “Directions for improvements”).

### **Strengths**
* The paper is very well written and presented. Results are reported in a systematic and clear manner.
* The experimental evaluation is rigorously conducted: the authors always verify the statistical significance of any observed performance gap, a practice that should definitely be adopted more broadly in our community.
* IGEL is discussed completely, in particular w.r.t. its limitations.
* Casting IGEL as a variant of the 1-WL test enables an easier and more immediate understanding of its functioning, especially on certain graph families sharing regularities in their topology. For example, this facilitates comprehending the limitations of IGEL over families of SR graphs.
* IGEL finds application in virtually all Graph Representation Learning algorithms and is generally domain-agnostic, making it a flexible approach of potential wide-spread application.

### **Weaknesses**

* Contextualisation and relation with previous works
    * The authors miss to position IGEL w.r.t. relevant previous work. There are, in particular, two research directions which relates to IGEL, and for which it would be interesting, if not important, to comparatively discuss the proposed approach:
        * There is vast literature on node positional encodings and node embeddings that relates to IGEL in that these auxiliary node representations can be seamlessly plugged into downstream models for graph learning tasks. Examples include RandomWalk and Laplacian positional encodings as well as DeepWalk, Node2Vec and TransE embeddings. The expressiveness of these approaches is not well characterised, but it is known that some of them allow to disambiguate between 1-WL equivalent graphs. Some others may lose equivariance, but this does not hinder their application to e.g. node-wise tasks. Positional encodings are also used in the context of graph classification in graph transformer models.
        * Some recent, provably powerful approaches such as GNN-AK(+) [1], NGNN [2], DEA-GNNs [3] leverage the same intuition to go beyond 1-WL, that is the use of ego-networks and / or node distance information. These methods are generally end-to-end, in the sense that they employ this form of information directly in the message-passing process. What is the advantage of IGEL w.r.t. these ones?
        * The work in [1] is of particular relevance: it describes a model, GNN-AK+, which jointly leverages both ego-networks and distance information and possibly resembles a neural counterpart of IGEL. Additionally, similarly as the present work, it also proposes a variant of the WL algorithm which directly encodes ego-network subgraph structures. The authors cite this work, but no explicit comparison is discussed.
* Experimental benchmarking
    * Whilst the result analyses are statistically rigorous, the real-word experiments:
        * Target relatively small benchmarks (Enzymes, Mutual, Proteins, PTC) — Nowadays there exist several larger scale publicly available datasets from a variety of domains, available from the Open Graph Benchmark; the authors do not benchmark IGEL on any of them.
        * Do not to benchmark against some related baselines — As the authors present IGEL as a preprocessing step generating discriminative node features based on ego-networks and node distances, it would be fairly reasonable to directly compare IGEL-augmented models with other strategies for positional encodings (Random-Walk-, Laplacian- or Matrix-Factorisation-based) or expressive GNNs based on ego-networks and distance information (GNN-AK(+) [1], NGNN [2], DEA-GNNs [3]).
    * No details on the experimental setting and procedure is reported, including how models are trained and evaluated.
* Form and syntax
    * There appear to be some inconsistency between American and British spelling, e.g. “stabilizes” (American), “neighbours” (British)
    * “e.g.” is often used in place of “i.e.” but the two Latin acronyms have two different meanings (“exempli gratia” vs. “id est”)
    * The authors sometimes make confusion between the image computed by a function for a specific input and the codomain the function is defined onto. e.g. “The output of 1-WL is [in?] $\mathbb{N}^n$” (line 50). They also use the symbol $\mapsto$ when defining a function in terms of its domain and codomain, where $\rightarrow$ would instead be more appropriate.
    * Some indexing typos are found in the appendix, amongst others: “Let G1 and G2 be two d-regular graphs such that |V1| = |V1|”.

### **Questions**
* The time complexity analyses does not seem to explicitly account for the complexity involved in the extraction of ego-networks, or otherwise the computation of the ego-network degrees $d_{\mathcal{E}^\alpha_G}(v)$ and set of neighbours at a specified distance from the root (line 5, Algorithm 2). In other words, the analysis accounts for the cardinality of multisets (lines 4–5 in Algorithm 2) but not the computation required to construct them. Can the authors clarify this aspect? Would the overall asymptotic complexity change?
* Can the authors clarify the sentence at lines 119-120? I am referring to: “and the ESAN framework when leveraging ego-networks with root-node flags as a subgraph sampling policy (EGO+) [19], which is as powerful as the 3-WL.”. ESAN comprises different possible choices for base encoder architectures, and the reported claim may not hold in some cases. For example, to the best of the reviewer’s knowledge, DS-GNN with EGO+ policy and MPNN base encoder is not yet generally known to be lower-bounded by 3-WL.
* (Minor) In the proof of Remark 2, lines 325 through 327 the authors may want to specify that $\beta > 0$ for the result to hold? Or alternatively, it may also be specified that there are no non-trivial SR graphs with both $\beta = 0, \gamma = 0$?

### **Directions for improvement**
* The paper would greatly benefit from a better contextualisation of IGEL w.r.t. (1) other works proposing node-embeddings or positional encodings and (2) expressive GNNs employing ego-networks and distance information. A more articulated comparative discussion would be really interesting and relevant for readers (invariance, complexity, expressiveness, applicability). Some experimental comparison would be important as well. For example, the authors could benchmark IGEL against GNN-AK+ and laplacian encodings on larger graph- and node-wise benchmarks from OGB (https://ogb.stanford.edu/) as well as subgraph counting (GNN-AK+ has already been tested on this task).
* The authors do not explicitly discuss and prove the relation between IGEL and 1-WL, other than exhibiting a pair of co-spectral (regular) graphs that are distinguished by IGEL but not by 1-WL. I would definitely include a _formal proof_ showing that IGEL is strictly more expressive than 1-WL (is it?). This can be articulated in two steps: (1) show that, under certain conditions, IGEL is at least as expressive as 1-WL: any graph pair separated by 1-WL is also separated by IGEL (2) show there exists pairs distinguished by IGEL but not by 1-WL — one potential example is reported by the authors, but they would probably need to formalise this into a lemma to support the overall proof.
* To also benchmark IGEL on larger scale graph learning benchmark would be a great addition to the presented work.
* The authors could discuss whether IGEL is permutation equivariant. This analysis is particularly important in the field of expressiveness of graph learning methods. It is my intuition that IGEL is, in fact, equivariant, and the authors could try to prove this formally.
* Remarks 2 and 3 in the Appendix follow the proof where they are referred to. I would definitely move them upwards, and potentially rename to “Proposition”s or “Lemma”s.
* Formal and syntactic inconsistencies and imprecisions can be addressed (see above).

### **References**
[1] Zhao et al., "From Stars to Subgraphs: Uplifting Any GNN with Local Structure Awareness", 2022

[2] Zhang & Li, "Nested Graph Neural Networks", 2021

[3] Li et al., "Distance Encoding: Design Provably More Powerful Neural Networks for Graph Representation Learning", 2020

---

### Official Review · Reviewer_aKV8 · 2022-10-19

**Overall Score:** 6
**Confidence:** 4

**Review:**

Summary:
The paper presents a structural encoding method (IGEL) - Section 2 - to produce node structural features that is based on ego-networks of the nodes. Such structural features can be pre-computed and can be used as additional node features with any message passing GNN. Theoretical analyses - Section 3.1 - show that IGEL enhances the expressivity power of the underlying MP-GNN as the encoding it provides is powerful than 1-WL in terms of distinguishing non-isomorphic graphs. Experimental studies - Section 3.2 - on multiple synthetic and real world graph benchmarks show that IGEL generally improves the performance when augmented on an MP-GNN or non-GNN method such as MLP.

To contextualize from one perspective (among others), this paper's contribution, IGEL, aligns with recent proposals in MP-GNN to augment node features with 'structural or positional' encodings that can be pre-computed by using the available graph structure. As with existing such methods, IGEL does not increase the complexity of the MP-GNN used (as mentioned in line 38: without modifying model architectures), while improves the expressivity.


Strengths:
1. As pointed in the summary, the paper's contribution is a simple but powerful method to improve any GNN without the architecture's modification, but by only using pre-computed feature vectors for every node.
2. The paper presents a balance of both theoretical and experimental justification of why IGEL is a useful technique. From theoretical perspective, the paper shows enhancement using example cases as well as synthetic graph tasks, while on the experimental side, empirical enhancements are shown on link, node as well as graph level tasks.
3. The expressivity upper bounds (Thm. 1) of IGEL has been included as part of the theoretical analyses since it addresses readers curious questions of the limits of IGEL as it is based on derived local ego-networks of nodes.
4. IGEL is able to model graph structural information of a node and its neighbors presence at certain hops (alpha parameter), along with degree information. This information seems key to IGEL encoding information that is not otherwise modelled by 1-WL.

Weaknesses:
While there seem no noticeable weakness in writing of the paper, following few things require clarification:
1. The representation of the multi set as node feature vectors could seem unclear in lines 65-66. Although the Fig. 1 demonstration is helpful, the vector inside Fig. 1 is not described as well as perhaps the sentence 'where the i-th index contains the frequency of path-length and degree pairs (λ, δ)' could be made clear as the 'i' is not described.
2. In the experiments, the alpha distance used does not seem to be mentioned for all the benchmarks. How is the alpha parameter chosen? What is the effect of varying alpha? How can it intuitively help a GNN or a node look at its longer-distanced neighbhors?
3. Additional weaknesses have been mentioned under the 'Feedback' heading below.
- Minor issue: line 51 'test' and 'tests' are repeated. Or perhaps the paper meant (denoted k-WL tests) in place of (denoted k-WL)?

Feedback:
1. The paper shows empirical results that show IGEL enhances the underlying GNN, as well as in some cases, comparison with k-hop, GSN, ESAN. It is of curious question on how IGEL compares directly with recent positional or structural encoding methods as I believe IGEL falls under that category which is also mentioned in the paper that IGEL is a pre-processing step to derive node attributes based on graph structure. In this case, GSN is one method that has directly been compared in Table 2, which shows that structural encoding of GSN is always better than the proposed structural encoding - IGEL. An experiment of fixing a MP-GNN and varying structural encodings would help to answer this question and also address one of the the paper's initial motivations (line 35-36).
2. Extracting ego-network information to improve GNN does not see new. In the literature, there exist works that leverage nodes' ego networks to derive higher-order structural information, eg. Sandfelder, D. et al., Ego-gnns: Exploiting ego structures in graph neural networks. 2021. How does IGEL compares with such works in the literature is missing in the paper.
3. The experimental benchmarks seem small and perhaps recent tasks on OGB can be used to compare with recent GNN works.


Based on the above comments, I understand the paper's contribution is elegant and could be useful to improve GNN being architecture-agnostic. To the best of my knowledge, there exist some ego-networks based proposals in literature to improve GNNs in the literature which modify the underlying GNN architecture. In this paper, the use of ego-network information seem relatively elegant and the paper also shows enhancements brought by it.
At the same time, there are a few issues that can be clarified as mentioned in Weaknesses above, and if possible (I understand all feedback would be difficult to address in a limited time frame), some questions in the Feedback section can be addressed which are of direct concern to IGEL. I hence recommend for a weak accept.

---

### Official Review · Reviewer_LGXf · 2022-10-20

**Overall Score:** 6
**Confidence:** 4

**Review:**

**Summary**: In this work, the authors propose IGEL, a newly designed structural encoding to produce features that boost classic Message Passing Neural Networks (MPNNs) beyond 1-WL expressivity. The authors theoretically analyze the relation between IGEL and 1-WL test, and give expressivity upper bounds on the proposed IGEL. Besides, experiments on graph/node/link classification, graph isomorphism detection, and graphlet counting are conducted to show the empirical performance of the IGEL.

**Strengths**:
- The idea of using sparse vectors to encode typical structural information (e.g. degrees) of ego-networks as input features is interesting.
- The authors provide expressivity upper bounds on the proposed IGEL.
- The experiments cover several typical graph learning tasks.

**Weaknesses**:

**Regarding the relation between IGEL and Subgraph GNNs**: The procedure of compute the IGEL encoding is (1) for each node, generate its k-hop ego-network (with k being pre-defined hyperparameters.) (2) for nodes in the induced ego-network, they are partitioned into different sets according to the shortest path distance between them and the corresponding node on the induced ego-network. (3) for each node, the degree information is encoded. For Subgraph GNNs like ESAN [1], the first step of the IGEL exactly corresponds to the subgraph generation. Besides, as pointed out in a concurrent work [2], the ESAN implicitly encodes the pairwise distance between nodes, and the degree information can also be easily extracted via aggregation. Thus, there exists a strong relationship between IGEL and Subgraph GNNs. The authors should clarify this to better position this work in the literature.

**Regarding the expressivity between IGEL and 1-WL test**: The authors show that there exist $d$-regular graphs that can be distinguished by IGEL while they cannot be distinguished by 1-WL test. However, the authors should also show that for any pair of graphs that are distinguished by 1-WL test, the IGEL can distinguish them as well. Although it may seem to be easy to prove, the authors should provide such analysis for the completeness of the claims.


**Minor Comments**
- Eq.(2) is confusing.
- In Line 55, why do perturbations produce different node-level representations for 1-WL?
- The performance gain of the proposed IGEL is minor on TU datasets compared to baselines. Besides, it is also reasonable to further verify the IGEL on datasets with larger scale (e.g., OGB [3] and Benchmarking GNNs [4]).

**Overall**, if the authors can address my above concerns well, I would like to increase my scores accordingly.

******** After Rebuttal ******** I lean towards acceptance. The authors should consider including the above advice in the full version of this paper.

[1] Bevilacqua B, Frasca F, Lim D, et al. Equivariant Subgraph Aggregation Networks[C]//International Conference on Learning Representations. 2021.

[2] anonymous. Rethinking the Expressive Power of GNNs via Graph Biconnectivity. https://openreview.net/forum?id=r9hNv76KoT3

[3] Hu W, Fey M, Zitnik M, et al. Open graph benchmark: Datasets for machine learning on graphs[J]. Advances in neural information processing systems, 2020, 33: 22118-22133.

[4] Dwivedi V P, Joshi C K, Laurent T, et al. Benchmarking graph neural networks[J]. arXiv preprint arXiv:2003.00982, 2020.

---

### Official Review · Reviewer_CP3S · 2022-10-22

**Overall Score:** 6
**Confidence:** 4

**Review:**

**Extended Abstract Summary:**

The authors propose IGEL, which augments information about local ego network knowledge through preprocessing to standard message passing GNNs to improve their expressivity over 1-WL.  The authors provide supporting theory and corroborative experiments on real world and synthetic graph datasets to validate their proposal.

**Strengths:**
1. The idea to use sparse vectors and encode ego network knowledge into the network is interesting
2. Good experimental results on graph datasets
3. Authors explicitly show the limitation/ inability of IGEL to be unable to work on strongly regular graphs.

**Weaknesses & Corresponding Questions & Suggestions:**
1. Lack of appropriate comparison/ contrast with prior works:  The work doesn't appropriately compare with explicit Ego GNN methods [1], [4] or with subgraph GNNs [2,3,5] - Almost all of which would be able to implicitly learn the sparse vector encoding used here.
2. Lack of novelty: Ego network extraction is not new [1,4] - and unfortunately, has not been acknowledge by the authors.
3. Lack of details about parameter selection e.g. $\alpha$ - what properties of the graph is this reliant on?

**Minor:**
1. The proof of comparison with 1-WL is incomplete - doesn't show capable of distinguishing everything that 1-WL can

**References:**
1. You, Jiaxuan, et al. "Identity-aware graph neural networks." Proceedings of the AAAI Conference on Artificial Intelligence. Vol. 35. No. 12. 2021.
2. Zhang, Muhan, and Pan Li. "Nested graph neural networks." Advances in Neural Information Processing Systems 34 (2021): 15734-15747.
3. Zhao, Lingxiao, et al. "From stars to subgraphs: Uplifting any GNN with local structure awareness." arXiv preprint arXiv:2110.03753 (2021).
4. Sandfelder, Dylan, Priyesh Vijayan, and William L. Hamilton. "Ego-gnns: Exploiting ego structures in graph neural networks." ICASSP 2021-2021 IEEE International Conference on Acoustics, Speech and Signal Processing (ICASSP). IEEE, 2021.
5. Bevilacqua, Beatrice, et al. "Equivariant subgraph aggregation networks." arXiv preprint arXiv:2110.02910 (2021).

---

### Meta-Review · Area_Chair_4jnp · 2022-11-09

**Confidence:** 3
**Recommendation:** Accept

**Meta Review:**

The paper introduces an ego-net-based preprocessing routine to provably enhance the expressivity of standard GNNs. limited by the 1-WL. All reviewers criticize that the idea is not entirely novel and the relationship to other works, especially subgraph-based GNNs, should be clarified. Moreover, some parts are not entirely clear. However, all reviewers liked the simplicity of the approach and the promising experimental results.

I advise the authors to thoroughly revise the paper.

---

### Decision · Program_Chairs · 2022-11-22

Accept (Poster)